# The Comparative Virulence of *Francisella tularensis* Subsp. *mediasiatica* for Vaccinated Laboratory Animals

**DOI:** 10.3390/microorganisms8091403

**Published:** 2020-09-12

**Authors:** Vitalii Timofeev, Galina Titareva, Irina Bahtejeva, Tatiana Kombarova, Tatiana Kravchenko, Alexander Mokrievich, Ivan Dyatlov

**Affiliations:** State Research Center for Applied Microbiology and Biotechnology (SRCAMB), Obolensk 142279, Russia; titarevag@mail.ru (G.T.); bahtejeva@mail.ru (I.B.); kombarova@obolensk.org (T.K.); tbkrav@mail.ru (T.K.); mokrievich@obolensk.org (A.M.); Dyatlov@obolensk.org (I.D.)

**Keywords:** *Francisella tularensis*, subsp. *mediasiatica*, vaccination, virulence

## Abstract

Tularemia is a severe infectious disease caused by the Gram-negative bacteria *Fracisella tularensis*. There are four subspecies of *F.*
*tularensis*: *holarctica*, *tularensis*, *mediasiatica,* and *novicida*, which differ in their virulence and geographic distribution. One of them, subsp. *mediasiatica* remains extremely poorly studied, primarily due to the fact that it is found only in the sparsely populated regions of Central Asia and Russia. In particular there is little information in the literature on the virulence and pathogenicity of subsp. *mediasiatica*. In the present article, we evaluated the comparative virulence of subsp. *mediasiatica* in vaccinated laboratory animals which we infected with virulent strains: subsp. *mediasiatica* 678, subsp. *holarctica* 503, and subsp. *tularensis* SCHU within 60 to 180 days after vaccination. We found that subsp. *mediasiatica* is comparable in pathogenicity in mice with subsp. *tularensis* and in guinea pigs with subsp. *holarctica*. We also found that the live vaccine does not fully protect mice from subsp. *mediasiatica* but completely protects guinea pigs for at least six months. In general, our data suggest that subsp. *mediasiatica* occupies an intermediate position in virulence between spp. *tularensis* and *holarctica*.

## 1. Introduction

*Francisella tularensis*, a non-spore forming, encapsulated Gram-negative coccobacillus, is the etiologic agent of the potentially fatal zoonotic disease, tularemia. Currently, *F*. *tularensis* is divided into four subspecies: *tularensis* (*nearctica*), *holarctica* (*palaearctica*), *mediasiatica*, and *novicida*, which differ in their distribution and virulence in humans. While subsp. *tularensis* (mainly found in North America) and subsp. *holarctica* (widespread in the Northern hemisphere) are highly virulent for humans and can cause death among patients who do not receive timely and adequate treatment [1,2,3,4,5], subsp. *novicida* (present in North America, but some strains were isolated in tropical Australia and Southeast Asia (clinical isolates only)) is almost avirulent in humans, with only a few reports on its isolation in patients [6,7].

The fourth subspecies, subsp. *mediasiatica*, remains the least studied. For a long time, this subspecies was thought to be found only in some regions of Kazakhstan and Turkmenistan in Central Asia. This geographic location is responsible for the subspecies name (in Russian, “Middle Asia” is a synonym for “Central Asia”). Previously, very little was known about the virulence of this subspecies, with no registered cases of subsp. *mediasiatica*-caused tularemia among humans, although some data for animals show that its virulence in hares is similar to the virulence of subsp. *holarctica* [1]. In general, this subspecies has remained outside the area of interest of most researchers, due to its low prevalence. However, in 2013, we found subsp. *mediasiatica* strains in the Altai region of Russia [8]. We showed that the Altaic population of *F*. *tularensis* subsp. *mediasiatica* is genetically distinct from the classical Central Asian population and is likely endemic to Southern Siberia.

Although no cases of human subsp. *mediasiatica* causing infection have been detected in the Altai, the detection of this strain raised not only the theoretical question of the virulence of this subspecies but also an applied question: Are current antiepidemic measures (primarily vaccination) effective against this subspecies? For vaccination against tularemia, live vaccine strains are commonly used: LVS and 15 NIIEG (in Russia, NIIEG is the “Research Institute of Epidemiology and Hygiene”, now called the “Research Institute of Microbiology of Ministry of Defense of the Russian Federation”, where this vaccine was developed). Both these strains belong to the subsp. *holarctica*. The effectiveness of live vaccines for infection prevention, including infections caused by strains of the most dangerous subspecies, *tularensis*, has long been a focus of researchers. A number of publications have investigated how effectively existing vaccines can avoid infection in various ways [9]. At the same time, the issue of overcoming post-vaccination immunity by strains of the subsp. *mediasiatica* has not been studied at all. This is logical since such strains are practically absent in the collections of microorganisms. However, there are already a few dozen such strains in the SRCAMB collection (the state collection of pathogenic microorganisms and cell cultures, Obolensk, Russia), and every year, several new strains are found in Altai. Therefore, we have both the reason and the opportunity to investigate the comparative pathogenicity of subsp. *mediasiatica* and its ability to cope with post-vaccinal immunity. Previously, we performed some preliminary experiments [10] showing that mice vaccinated with a live vaccine strain endured the infections caused by different subspecies in different ways. In this model, subsp. *mediasiatica* was shown to be less pathogenic than subsp. *tularensis* but more pathogenic than subsp. *holarctica*. However, the question remains whether the effectiveness of vaccination decreases over time. In the present manuscript, we investigate the dynamics of the pathogenicity of different subspecies in vaccinated mice and guinea pigs over six months.

## 2. Materials and Methods

### 2.1. Strains

The strains of *F*. *tularensis* used in the present study are listed in Table 1. All used strains were deposited in the SRCAMB collection. Two of the three subsp. *mediasiatica* strains—678 and 554—were isolated in the Altai region, while strain 120 was isolated in Central Asia.

### 2.2. Bacterial Cultures

The strains were grown at 37 °C on solid (FT-agar) and liquid (FT-broth) nutrient media (SRCAMB, Obolensk, Russia).

### 2.3. Animal Experiments

#### 2.3.1. Ethics Statement

All protocols for animal experiments were approved by the State Research Center for Applied Microbiology and Biotechnology Bioethics Committee (Permit No: VP-2019/2). They were performed in compliance with NIH Animal Welfare Insurance #A5476-01 issued on 02/07/2007 and the European Union guidelines and regulations on the handling, care, and protection of laboratory animals (https://eur-lex.europa.eu/eli/dir/2010/63/oj).

A minimum number of animals was used for the experiments. The approved protocols provided scientifically validated humane endpoints, including pre-set criteria for the euthanasia of moribund mice via CO_2_ inhalation. In our study, the animals were euthanized when they became lethargic, dehydrated, moribund, unable to rise, or non-responsive to touch. The health conditions of the animals were monitored at least twice a day.

#### 2.3.2. Animals

BALB/C mice (5–8 weeks-old, 18–20 g) and guinea pigs (5–8 weeks-old, 275 ± 25 g) of both genders, purchased from the Laboratory Animals Breeding Center Shemyakin and Ovchinnikov Institute of Bioorganic Chemistry, Russia, were used in the experiments.

Animals were housed in polycarbonate cages with space for comfortable movement (5 mice in a 484 cm^2^-cage and 3 guinea pigs in a 864 cm^2^-cage) with ad libitum access to food (Mouse Mixed Fodder PK-120, and Guinea Pig Mixed Fodder КК–122, Laboratorkorm, Russia) and tap water, under constant temperature and humidity conditions (22 °C ± 2 °C and 50% ± 10%, respectively) and a 12-h light/12-h dark cycle.

### 2.4. Vaccination

Animals were vaccinated subcutaneously with *F*. *tularensis* subsp. *holarctica* 15 NIIEG cells in PBS (25 CFU per mouse and 5 × 10^4^ CFU per guinea pig) in the inner part of the upper thigh.

Vaccination was carried out while considering the time-point, such that by the time of the simultaneous infection of the animals, we had groups of mice vaccinated 30, 60, 90, 120, and 180 days prior and groups of guinea pigs vaccinated 30, 90, and 180 days prior.

### 2.5. Pathogenicity Determination

Within the above terms after vaccination, groups of 25–52 mice and 3–5 guinea pigs were challenged with virulent strains of all three studied subspecies of *F*. *tularensis*—*holarctica, tularensis* and *mediasiatica* (1000 CFU per animal). The animals were then inoculated subcutaneously with 0.1 mL *F*. *tularensis* cells in PBS.

We observed the states of the animals for 21 days after infection, during which we registered their deaths. The animals’ body weight changes were registered every 2–4 days for 16 days after infection. Mice were weighed in groups of five animals (or less in the event of a death), and the guinea pigs were weighed individually. The average weight changes were calculated compared to each animal’s body weight on the day of infection. Dead animals were autopsied and subjected to bacteriological studies.

### 2.6. Statistics

The obtained data were statistically processed using the GraphPad Prism 7 program (https://www.graphpad.com/). The data were presented as the means ± SEM. For comparing the weight and lifespan data, while one-way ANOVA (Mann–Whitney), two-way ANOVA (Dunnett’s multiple comparisons), and nonparametric (Kolmogorov–Smirnov and Kruskal–Wallis) tests were used to evaluate the significance between groups. For the survival curve, a log-rank test, a log-rank test for trends and a Gehan–Breslow–Wilcoxon test were used. P values less than 0.05 were considered significant.

## 3. Results

### 3.1. Mice Biomodel

To evaluate the comparative ability of subsp. *mediasiatica* to overcome artificial active anti-tularemia immunity, we used mice immunized with the vaccine strain *F. tularensis* ssp. *holarctica* 15 NIIEG (see materials and methods) at 30, 60, 90, 120, and 180 days before the experiment. We infected the mice with three virulent strains of *F. tularensis*: subsp. *holarctica* 503, subsp. *tularensis* SCHU, and subsp. *mediasiatica* 678 (from 25 to 52 individuals per group, with a total of 478 mice). The control groups of unvaccinated mice died without exception, regardless of the infecting strain. For convenience of presentation, these groups are referred to below as the 503 group, the 678 group, and the SCHU group. The observation results are presented in general terms in Table 2.

In more detail, the obtained results are described below for each time point in the order of the increasing time interval between vaccination and infection, as illustrated in Figure 1, Figure 2, Figure 3, Figure 4 and Figure 5.

At 30 days between immunization and infection. Despite several recorded deaths, at this time-point, we did not observe any statistically significant differences between all three groups of mice; vaccination effectively protected the mice from infection with all tested strains (Figure 1).

At 60 days between immunization and infection. At this time, the differences between the infecting strains began to appear. Again, group 503 survived completely, unlike the 678 group and SCHU (Figure 2). The highest mortality rate was in the SCHU group (21.2%) (*p*_503-SCHU_ = 0.0157). In addition, this group had the shortest life expectancy. The mortality rate in the 678 group was 11.4% (*p*_503–678_ = 0.084, not significantly different). There was no statistically significant difference between the groups in weight loss caused by the disease.

At 90 days between immunization and infection. At this time-point, the differences between groups became more noticeable (Figure 3). The mortality rate in group 503 remained 0%, but that in the 678 group reached 27.6%, with 40.7% observed in the SCHU group (*p*_503-SCHU_ = 0.0009, *p*_503–678_ = 0.0085). The average lifespan in the 678 group was slightly lower than that in the SCHU group. We observed the lowest significant body weight loss in group 503 compared to that in the other two groups (*p* < 0.05).

At 120 days between immunization and infection. At this point in time, the results were generally similar to those of the previous time-point (90 days). The mortality rate in group 503 was 6.3%, with 25% in the 678 group and 55.6% in the SCHU group (*p*_503-SCHU_ = 0.0027, *p*_503-678_ = 0.0176). Here, statistically significant differences between the three groups in terms of lifespan disappeared. Body weight loss was also significantly higher in the 678 and SCHU groups compared to that in group 503 (Figure 4).

Notably, at exactly this point, we observed statistically significant differences between subsp. *mediasiatica* and subsp. *tularensis*—the mortality rate in the SCHU group was significantly higher than that in the 678 group (*p* < 0.05).

At 180 days between immunization and infection. The mortality rate reached a maximum in all groups, with 13.3% in the 503 group, 40% in the 678 group, and 57.7% in the SCHU group (*p*_503-SCHU_ = 0.003, *p*_503-678_ = 0.043). The average lifespan of the mice after infection also decreased; the average lifespan in the 678 and SCHU groups was lower than that in the 503 group. At the same time, in contrast to the 90- and 120-day terms between immunization and infection, we did not observe any reliable differences in body weight during illness caused by different strains. In addition, contrary to expectations, we did not find any statistically significant differences between the 678 group and SCHU group, unlike the results for the term of 120 days.

As a result, a certain trend was observable from the 60th day after vaccination onwards. As the time from vaccination to infection increased, the severity of tularemia caused by infection also increased. Figure 6 illustrates the increase in mortality rate as the time between vaccination and challenge increased for all three strains used.

The tested strains were clearly divided into two groups according to the severity of the induced disease, with strain 503 causing disease of “moderate” severity and strains 678 and SCHU causing a much more severe disease.

To confirm this finding, we conducted an additional experiment using six more strains, two of each from among the three subspecies: subsp. *holarctica* 1045 and X-3; subsp. *mediasiatica* 120 and 554 and subsp. *tularensis* 8859 and ACole (Table 1).

We infected the mice vaccinated 90 days prior with these strains (177 mice in total, with 27 to 32 mice in each group), and recorded only their deaths, without measuring body weight. After 21 days of observation, we obtained the results shown in Table 3 and Figure 7.

Based on the data obtained, the trends that we observed earlier remained fully applicable. Mice infected with subsp. *holarctica* strains had the lowest mortality rates, while infection was most severe in mice infected with subsp. *tularensis* strains. Subsp. *mediasiatica* caused a significantly more severe disease than that caused by subsp. *holarctica* strains. At the same time, the difference with subsp. *tularensis* was unreliable.

### 3.2. Guinea Pig Biomodel

We used guinea pigs (*n* = 29) immunized 30, 90, and 180 days before infection (three to five guinea pigs per group) with strains 503, SCHU, and 678. We observed these guinea pigs and recorded their deaths over 21 days. The results are presented in Table 4 and Figure 8. In our experiment, the guinea pigs infected with strains 503 and 678 did not die at all, regardless of the period between immunization and infection. At the same time, animals infected with the SCHU strain did not die only at 30 days post-immunization. With longer periods, all animals died without exception.

Additionally, we measured the dynamics of body weight for the group infected at the 90th day after vaccination. We found that the body weight loss was not very noticeable in groups 503 and 678, reaching no more than 4–6% on the 3rd–5th day post infection. At the same time, animals infected with the SCHU strain lost up to 25% of their body weight before death.

## 4. Discussion

The comparison of different non-attenuated strains of *F. tularensis* (including those belonging to different subspecies) according to their virulence using the direct infection of small laboratory animals is complicated by the high virulence of this pathogen: The LD_50_ value is only 10^0^–10^1^ CFU/animal [11,12]. This is also true for subsp. *mediasiatica* [8,10]. As shown in Table 1 (Materials and Methods section), the LD_50_ values for the strains of this subspecies do not generally differ from those of strains of other subspecies. To obtain a statistically reliable result with this approach, a large number of animals is required, which is difficult to justify in terms of the cost of the work, as well as for bioethical reasons. We tried to overcome this difficulty by using previously vaccinated animals. After vaccination, the animals acquired immunity that protected them from infection only from low virulent strains, whereas highly virulent strains were able to overcome this protection [11,12]. This made it possible to differentiate strains by virulence using even a relatively small number of animals in the experiment. Additionally, the challenge of vaccinated animals allowed an assessment of the effectiveness of the vaccination, demonstrating to what extent it was able to protect against that particular infecting strain. As we already mentioned for subsp. *mediasiatica*, this issue remains insufficiently studied and is, therefore, especially interesting. We already used vaccinated mice in our previous work [10], for which this approach seemed reasonable and allowed the *F. tularensis* subspecies to be arranged in decreasing order of virulence for mice in the following order: subsp. *tularensis*–subsp. > *mediasiatica*–subsp. > *holarctica*. In that work, only one preliminary experiment was carried out with the infection of mice 3 weeks after vaccination. However, post-vaccination immunity can eventually decrease [13,14]. Therefore, it remains interesting to trace how the pathogenicity of different subspecies changes with an increase in the time interval between vaccination and infection. In addition, different species of macroorganisms can react differently to infection and differ in their immune responses to vaccination. To draw conclusions about the comparative pathogenicity of certain microorganisms, including different subspecies of *F. tularensis*, it is highly desirable to use more than one species of laboratory animals [15,16,17]. Thus, in our present work, we significantly increased the duration of the experiment to 180 days and used an additional biomodel of guinea pigs.

For vaccinated mice at the beginning of the experiment, infection with any subspecies was shown not to be practically dangerous. However, with an increase in the time from vaccination to infection, the immune defense gradually became ineffective, and the mortality rate increased accordingly (Table 2 and Figure 1, Figure 2, Figure 3, Figure 4, Figure 5, Figure 6 and Figure 7). This decrease in effectiveness was directly proportional to time and depended on the subspecies. For subsp. *mediasiatica* and subsp. *tularensis,* the immune defense was overcome starting from 60 days. However, for subsp. *holarctica,* even at 180 days after vaccination, most of the mice survived.

The same trends were observed for bodyweight loss during illness: (1) Weight loss increased with an increase in the time between vaccination and infection; (2) mice infected with subsp. *mediasiatica* and subsp. *tularensis* strains lost more weight than the mice infected with subsp. *holarctica* strain, with the exception of groups infected on the 180th day after vaccination, which lost approximately the same weight, regardless of the infecting strain.

We should note the mice that died from infection during the experiments seemed to lose bodyweight faster and more significantly than those that recovered. Therefore, every death led to an increase in the average body weight in the group. Thus, weighing the mice in groups was an integral indicator that averaged the responses to the disease among the two different subgroups–dead and recovered. It would be better to carry out individual weighing, which would possibly be more informative, despite requiring much more labor. Secondly, the weight loss caused by infection and subsequent weight gain during recovery is a process with complex dynamics over time. For example, it is difficult to determine what represents an indicator of a more severe disease–a significant decrease in body weight and rapid recovery or a less significant weight loss that persists for a longer time? Therefore, to compare the strains by pathogenicity, we used only one indicator of body weight, which is the most illustrative and easily detectable in our opinion–the maximum value (as a percentage of the initial body weight on the day of infection) at which the average body weight of a mouse in a particular group decreased during the observation period.

In addition to the mortality rate and body weight dynamics, we sought to identify differences in the severity of infection caused by different strains using a measurement of the life expectancy of the mice that died from this infection. Contrary to our expectations, the lifespan did not differ regardless of the infecting strain and the time interval between vaccination and infection. There was only one exception: At 180 days, the mice who died of a strain 503 infection had lived significantly longer than the mice who died of strains SCHU and 678 (again, there was no significant difference in this indicator between these two strains). However, in the experiment with six additional strains the life expectancy during infection caused by subsp. *holarctica* strains was slightly less than that of the other two subspecies. Thus, the life expectancy during infection was shown to be an inconvenient and insufficient visual indicator of pathogenicity and cannot be used to compare the pathogenicity of different strains.

Nevertheless, both the mortality rate and bodyweight loss caused by infection indicate that subsp. *holarctica* strains had low pathogenicity for vaccinated mice, but the subsp. *mediasiatica* and subsp. *tularensis* strains were highly pathogenic. Here, a statistically significant difference between the subspecies *mediaasiatica* and *tularensis* appeared only for the strain ACole, which was shown to be the most pathogenic of the strains used in this work. However, when we averaged the results for all strains of one subspecies, the differences between subsp. *mediasiatica* and subsp. *tularensis* became statistically insignificant. Summarizing the results obtained using the vaccinated mouse model, we can conclude that subsp. *mediasiatica* is noticeably more pathogenic than subsp. *holarctica* and is comparable in pathogenicity to subsp. *tularensis*. Like subsp. *tularensis*, subsp. *mediasiatica* was able to overcome post-vaccinal immunity starting from 60 days after vaccination with a live vaccine strain. The reasons for this reduced ability of the vaccination to protect mice against infection with subsp. *mediasiatica* remain controversial. Two reasons for this phenomenon can be assumed: (1) the greater virulence of this subspecies provided by the greater specificity of its pathogenicity factors for molecular targets in the mouse organism and (2) the differences in the antigenic composition of different subspecies. In this case, the antibodies produced during vaccination with strain 15NIIEG belonging to subsp. *holarctica* did not effectively bind to the antigens of the strains of the two other subspecies, thus yielding a poorer degree of infection prevention.

It would be interesting to assess this result in mice vaccinated with attenuated strains belonging to subsp. *tularensis* and subsp. *mediasiatica* by subsequently infecting them with virulent strains of different subspecies. Vaccination with subsp. *mediasiatica* strains may be more effective in protecting against infection with virulent strains of both the same subspecies and subsp. *tularensis* (considering their similarity in pathogenicity demonstrated in the present work). However, at the moment, there is no attenuated subsp. *mediasiatica* strain that retains the ability to persist in animals for a long enough time for the formation of an immune response.

In the guinea pig model, we obtained very different results. The vaccinated guinea pigs were fully protected from infection with strains 503 and 678 throughout the entire period of the experiment (i.e., for 180 days, no single death was recorded) but not from the SCHU strain: all animals infected with this strain 90 days after vaccination (and latter) died.

Thus, vaccination reliably protected guinea pigs from infection caused by the subsp. *mediasiatica* strain, as well as infection with the subsp. *holarctica* strain.

However, we detected some differences between strains 503 and 678. During the autopsy of the guinea pigs, we observed that the animals that died from infection with the SСHU strain (90 or more days between vaccination and infection) had multiple lesions of the liver, spleen, and lymph nodes. Even the surviving animals infected with this strain (30 days between vaccination and infection) had enlarged inguinal lymph nodes (closest to the site of infection) and noticeable pathological changes in the spleen (spleens were enlarged and had multiple small abscesses). At the same time, the guinea pigs infected with strain 503 did not have such pathologies, regardless of the time elapsed between vaccination and infection. However, even though they all survived, the animals infected with strain 678 experienced some pathogenic changes. All of them had enlarged inguinal lymph nodes, and one guinea pig even had a necrotic lesion of this lymph node. Moreover, they all had enlarged spleens with small multiple abscesses, as in the animals infected with the SСHU strain at 30 days after vaccination. Thus, there is reason to believe that strain 678 causes a slightly more severe infection than strain 503, although this infection is neither fatal nor accompanied by greater weight loss during illness. However even taking these differences into account, for the vaccinated guinea pigs, subsp. *mediasiatica* was comparable in pathogenicity to subsp. *holarctica* but not to subsp. *tularensis* and was not able to overcome post-vaccinal immunity.

Thus, we have shown that subsp. *mediasiatica* is comparable in pathogenicity in mice with subsp. *tularensis* and in guinea pigs with subsp. *holarctica*. We also found that the live vaccine did not fully protect mice from subsp. *mediasiatica* but completely protected guinea pigs for at least six months. To date, there has been little information in the literature on the virulence and pathogenicity of subsp. *mediasiatica*. Articles that mention this subspecies (e.g., [18]) only reported it to be less virulent for animals than subsp. *tularensis* based on the primary source of this information (the works of Soviet scientists published in the 1960s–1970s). However, even in the original sources that we could find, information on the pathogenicity of subsp. *mediasiatica* was not detailed enough. For example, the authors in [19] only noted that “subsp. *mediasiatica* is slightly more virulent for domestic rabbits than subsp. *holarctica*”, without a detailed description of the experiments.

We originally studied the pathogenicity of subsp. *mediasiatica* for two biological models at once. However, we are aware that there is no reason to extrapolate our results to humans. It could be assumed that the virulence and pathogenicity of subsp. *mediasiatica* is even lower for humans than for guinea pigs, because subsp. *mediasiatica* strains have not caused a single known case of tularemia in humans. Moreover, since this subspecies was found in the Altai region of Russia in 2014, several of its strains have been isolated, but all of them were isolated only from ticks. There have been no clinical strains or strains isolated from captured wild rodents and/or from collected from corpses of rodents. Thus, our experimental data may not reflect the natural virulence of this subspecies in wild animals. However, these arguments remain speculative. Not a single purposeful expedition was conducted to collect subsp. *mediasiatica* strains to study the prevalence and natural virulence of this subspecies. All data were provided by the Altai anti-plague station, which cannot carry out such full-scale work. Thus, the lack of studies on field samples did not allow us to reconstruct the distribution of subsp. *mediasiatica* and its virulence among wild animals. Moreover, the absence of registered cases of tularemia caused by this subspecies in Russia may be due to insufficient diagnoses. In the mountainous Altai region, the infrastructure is poorly developed, and the population density is low, so the population is not sufficiently covered by medical supervision. Disease can occur in a secluded village, affect only isolated individuals, and/or circulate only in a mild form, thus not leading to the death or long-term disability of sick individuals. Such a case may remain outside the interest of doctors and not be registered. However even this probability suggests either a low prevalence of this microorganism in the Altai region or extremely insignificant clinical manifestations of the infection in humans sufficient to avoid the attention of doctors.

Nevertheless, we cannot completely exclude the hypothetical possibility of human tularemia caused by subsp. *mediasiatica* strains endemic to Russia. We must be prepared for such a possibility, and we must determine how to counteract it. Vaccination remains the primary way to avoid tularemia infection. In this way, the previously unexplored ability of existing antitularemia vaccines to prevent infection with subsp. *mediasiatica* has acquired new relevance. Although the results of our experiments were significantly different between the two different biomodels, in general, subsp. *mediasiatica* did not show an unexpectedly high ability to overcome post-vaccination immunity. However, this subspecies is pathogenic for vaccinated laboratory animals and is capable of causing a fatal infection or at least some pathological changes. This fact should be taken into account when developing new anti-tularemia vaccines. To assess the ability of such vaccines to prevent tularemia infection, it would be interesting to challenge vaccinated experimental animals with not only subsp. *holarctica* and subsp. *tularensis* strains, but also with subsp. *mediasiatica* strains.

Although subsp. *mediasiatica* has similar pathogenic properties to subsp. *tularensis* for mice and subsp. *holarctica* for guinea pigs, we can still say that, overall, subsp. *mediasiatica* assumes an intermediate position between the other two subspecies. Its similarity in virulence for mice with subsp. *tularensis* is most interesting from the perspective of the evolutionary relationship between *F. tularensis* subspecies, not from the perspective of epidemiology.

## Figures and Tables

**Figure 1 microorganisms-08-01403-f001:**
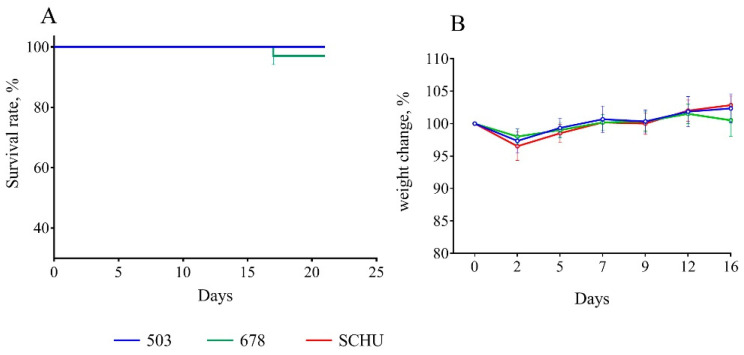
Survival rate (**A**) and bodyweight changes (**B**) of mice immunized 30 days before infection with virulent strains of *F. tularensis*: subsp. *holarctica* 503, subsp. *mediasiatica* 678, and subsp. *tularensis* SCHU. Values are indicated with the standard error of the mean (SEM) for each measurement point.

**Figure 2 microorganisms-08-01403-f002:**
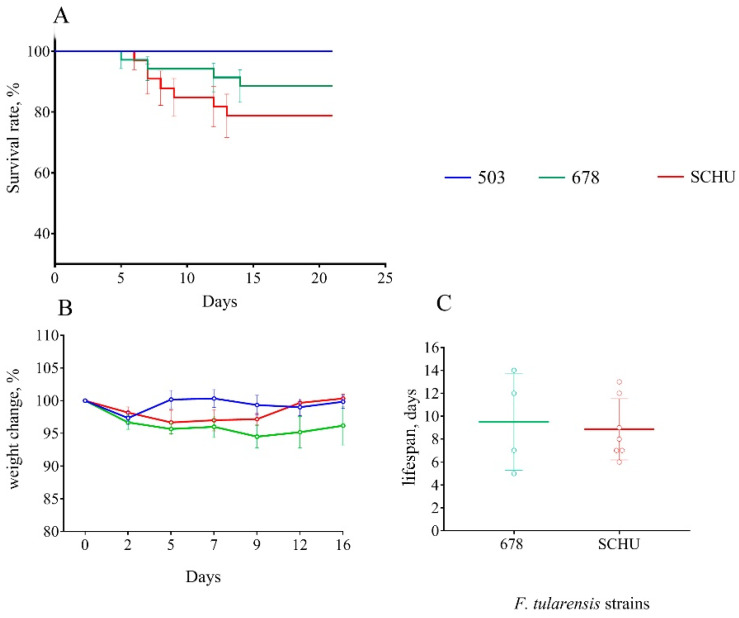
Survival rate (**A**), bodyweight changes (**B**), and lifespan (**C**) of mice immunized 60 days before infection with virulent strains of *F. tularensis*: subsp. *holarctica* 503, subsp. *mediasiatica* 678, and subsp. *tularensis* SCHU. Values are indicated with the standard error of the mean (SEM) for each measurement point.

**Figure 3 microorganisms-08-01403-f003:**
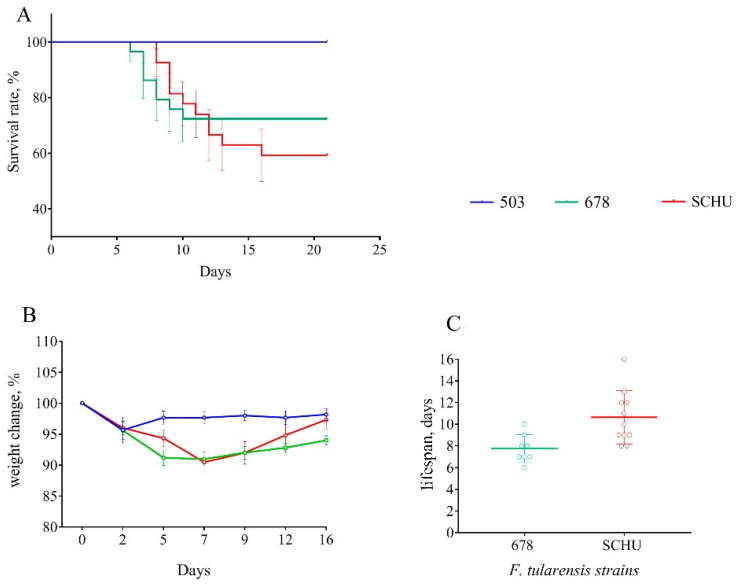
Survival rate (**A**), bodyweight changes (**B**), and lifespan (**C**) of mice immunized 90 days before infection with virulent strains of *F. tularensis*: subsp. *holarctica* 503, subsp. *mediasiatica* 678, and subsp. *tularensis* SCHU. Values are indicated with the standard error of the mean (SEM) for each measurement point.

**Figure 4 microorganisms-08-01403-f004:**
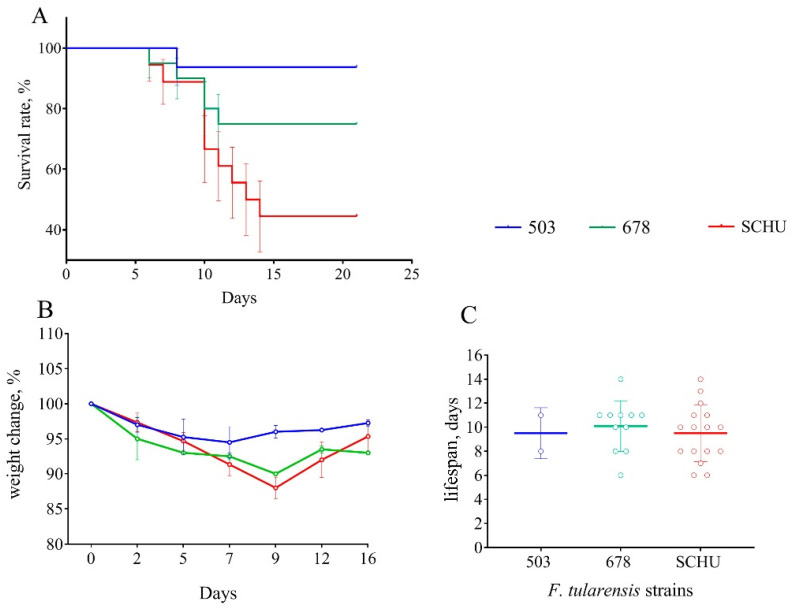
Survival rate (**A**), bodyweight changes (**B**), and lifespan (**C**) of mice immunized 120 days before infection with virulent strains of *F. tularensis*: subsp. *holarctica* 503, subsp. *mediasiatica* 678, and subsp. *tularensis* SCHU. Values are indicated with the standard error of the mean (SEM) for each measurement point.

**Figure 5 microorganisms-08-01403-f005:**
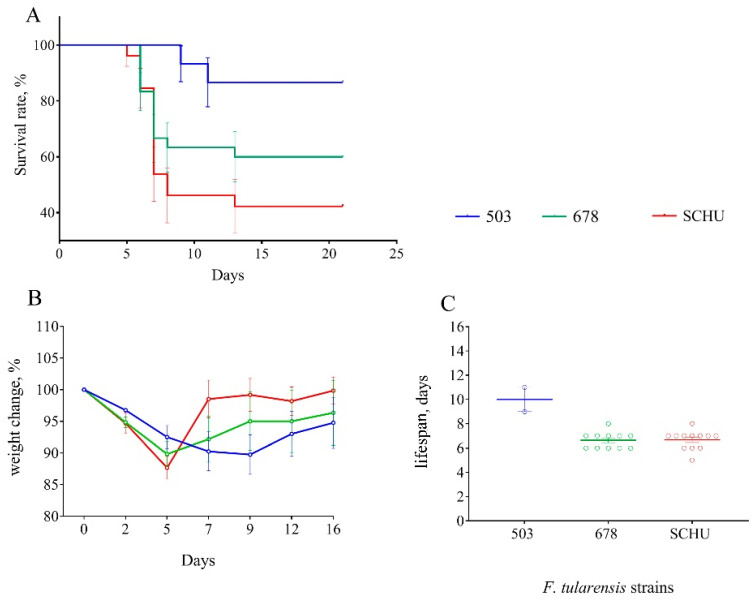
Survival rate (**A**), bodyweight changes (**B**), and lifespan (**C**) of mice immunized 180 days before infection with virulent strains of *F. tularensis*: subsp. *holarctica* 503, subsp. *mediasiatica* 678, and subsp. *tularensis* SCHU. Values are indicated with the standard error of the mean (SEM) for each measurement point.

**Figure 6 microorganisms-08-01403-f006:**
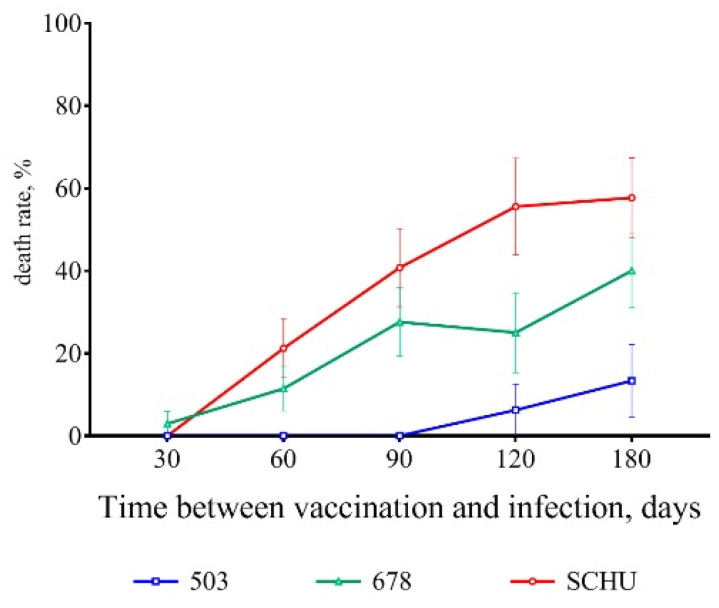
Dynamics of the mortality rate with an increase in the period between the vaccination of mice and their infection with virulent strains of *F. tularensis*: subsp. *holarctica* 503, subsp. *mediasiatica* 678, and subsp. *tularensis* SCHU. Values are indicated with the standard error of the mean (SEM) for each measurement point.

**Figure 7 microorganisms-08-01403-f007:**
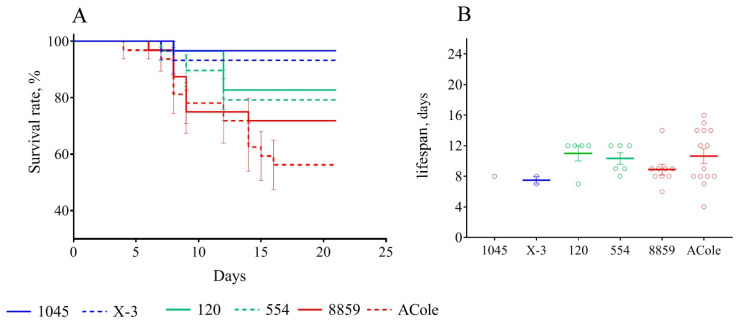
Survival rate (**A**) and lifespan (**B**) of mice immunized 90 days before infection with virulent strains of *F. tularensis*: subsp. *holarctica* 1045 and X-3, subsp. *mediasiatica* 554 and 120, and subsp. *tularensis* 8859 and ACole. Values are indicated with the standard error of the mean (SEM) for each measurement point.

**Figure 8 microorganisms-08-01403-f008:**
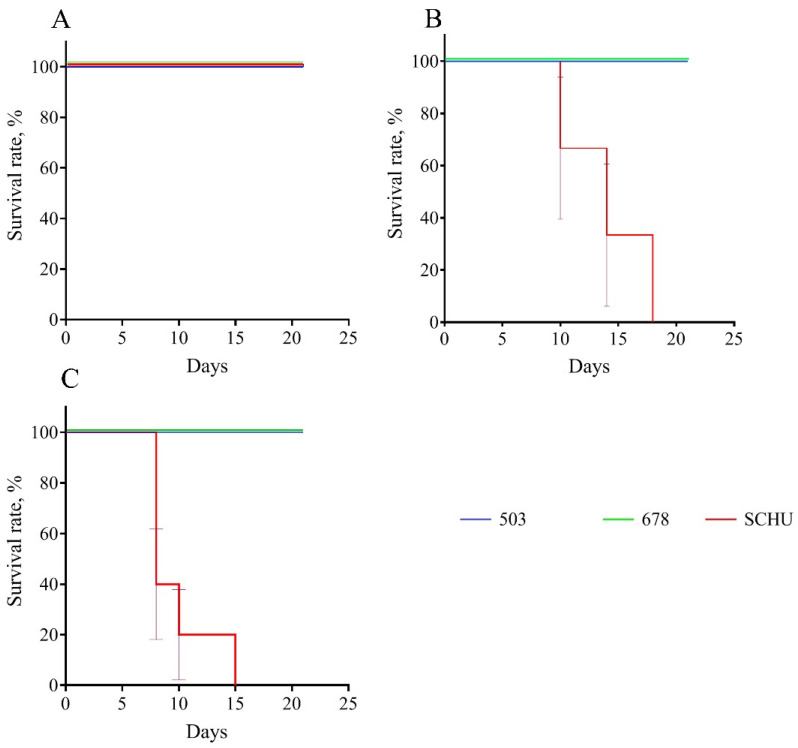
Survival rate for experimental tularemia in guinea pigs immunized 30 (**A**), 90 (**B**), and 180 (**C**) days before infection with virulent strains of *F. tularensis*: subsp. *holarctica* 503, subsp. *mediasiatica* 678, and subsp. *tularensis* SCHU.

**Table 1 microorganisms-08-01403-t001:** Strains of *F*. *tularensis* used in the present study.

Strain Name	Subspecies	LD_50_ for Mice, CFU/Animal	LD_50_ for Guinea Pigs, CFU/Animal
15NIIEG	*holarctica*	82	10^8^
503	*holarctica*	<5	<50
1045	*holarctica*	<5	<50
X-3	*holarctica*	<5	<50
SCHU	*tularensis*	<5	<50
8859	*tularensis*	<5	<50
ACole	*tularensis*	<5	<50
678	*mediasiatica*	<5	<50
120	*mediasiatica*	<5	<50
554	*mediasiatica*	<5	<50

**Table 2 microorganisms-08-01403-t002:** Mortality, lifespan, and body weight loss of immunized mice infected with virulent *F. tularensis* strains at different times after immunization.

Time between Vaccination and Infection in Days	Mortality Rate, %			Average Lifespan, Days			Body Weight Loss Maximum *, %		
	Group			Group			Group		
	503	678	SCHU	503	678	SCHU	503	678	SCHU
30	0	2.9 ± 2.9	0	No deaths recorded	17	No deaths recorded	2.7	2.0	3.5
60	0	11.4 ± 5.4	21.2 ± 7.1	No deaths recorded	9.5 ± 2.102	8.9 ± 1.01	2.7	5.5	3.3
90	0	27.6 ± 8.3	40.7 ± 9.5	No deaths recorded	7.8 ± 0.45	10.6 ± 0.74	4.3	9.0	9.5
120	6.3 ± 6.3	25 ± 9.7	55.6 ± 11.7	9.5 ± 1.5	10.09 ± 0.64	9.5 ± 0.59	7.2	17.7	12.7
180	13.3 ± 8.8	40 ± 8.9	57.7 ± 9.7	10 ± 1	6.64 ± 0.2	6.67 ± 0.22	10.3	10.2	12.3

* during observation period or until death.

**Table 3 microorganisms-08-01403-t003:** Mortality rate and lifespan of mice immunized 90 days before the experiment and infected with 6 virulent strains of *F. tularensis*. Values are indicated with the standard error of the mean (SEM).

Strain	Subspecies	Mortality Rate, %	Average Lifespan, Days
1045	*holarctica*	3.7 ± 3.6	8 *
X-3		7.4 ± 5	7.5 ± 0.5
120	*mediasiatica*	16.7 ± 6.8	11 ± 1
554		20.7 ± 7.5	10.3 ± 0.8
8859	*tularensis*	28.1 ± 7.9	8.9 ± 0.7
ACole		43.8 ± 8.8	10.6 ± 1

* Only one death was recorded, so it is impossible to indicate the SEM.

**Table 4 microorganisms-08-01403-t004:** Mortality and lifespan of immunized guinea pigs infected with virulent *F. tularensis* strains at different times after immunization. Values are indicated with the standard error of the mean (SEM).

Time Between Vaccination and Infection in Days	Strain	Mortality Rate, %	Average Lifespan, Days
30	503	0	No deaths recorded
	678	0	No deaths recorded
	SCHU	0	No deaths recorded
90	503	0	No deaths recorded
	678	0	No deaths recorded
	SCHU	100	14 ± 2.3
180	503	0	No deaths recorded
	678	0	No deaths recorded
	SCHU	100	9.8 ± 1.4

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
