# Peer review of "The Comparative Virulence of Francisella tularensis Subsp. mediasiatica for Vaccinated Laboratory Animals"

_microorganisms, 2020, doi:10.3390/microorganisms8091403_

Round 1
Reviewer 1 Report
Vitalii Timofeev et. al., in manuscript entitled „The comparative virulence of Francisella tularensis subsp. mediasiatica for vaccinated laboratory animals“valuated the virulence of subsp. mediasiatica in vaccinated laboratory animals in comparison with virulent strains: subsp. mediasiatica 678, subsp. holarctica 503, and subsp. tularensis SCHU within 60 to 180 days after vaccination.
Major commnets:
In all manuscript authors refer to virulence and pathogenicity although only mortality was follow in the experiment. For clearing that the strain is more or less virulent than other in animal models more studies should be taken including the bacterial load in the organs, histopathology etc.
- it would be interesting to see the bacterial load in the organs of animals that at least survived the infection. Authors, mention that results, but it should be added to the manuscript.
line 18- delete one that
Author Response
Dear Rewiever
We thank you for a careful assessment of our work, and very valuable criticisms. Please find below our point by point answers.
1) In all manuscript authors refer to virulence and pathogenicity although only mortality was follow in the experiment. For clearing that the strain is more or less virulent than other in animal models more studies should be taken including the bacterial load in the organs, histopathology etc. It would be interesting to see the bacterial load in the organs of animals that at least survived the infection. Authors, mention that results, but it should be added to the manuscript.
This is a fair point. We absolutely agree that the more data, the more interesting is the article. Definitely it would be interesting to track the dynamics of the bacterial load in the organs during the infectious process, the formation and the disappearance of histopathological changes, the changes in antibodies titers and in cellular markers of immunity etc.
But, to our deepest regret, the terms of the agreement in with present investigation was done, and its levels of funding did not allow to do such detailed research. The main question we had to answer was «Is there any reason to believe that subsp.mediasiatica, recently discovered in Russia, can overcome post-vaccination immunity».
Realizing that our data could be more complete, we nevertheless believe that even in the current form it may be of interest to researchers studying tularemia, and we hope that they will be sufficient for a thematic issue of "Epidemiology of Tularemia and Francisella tularensis "
2) line 18- delete one that
Done
Also you considered that «Moderate English changes required». English is not the native language of the authors, therefore, before submitting our manuscript, we ordered English editing at «MDPI's English editing service», and only after that, we submitted the edited manuscript to «Microorganisms». As you found the MDPI's English editing service inadequate, we contacted them again with your feedback, and manuscript was reedited. We hope that now the English language and style are fine
Reviewer 2 Report
The comments can be found in the attached file above.

Author Response
Dear Rewiever,
We thank you for a careful assessment of our work, and very valuable criticisms. Please find below our point by point answers.
1) Throughout the manuscript: It must be reviewed by a specialist with English as mother tongue, especially in the location of adjectives and substantives and for differentiating past tense and gerund
English is not the native language of the authors, therefore, before submitting our manuscript, we ordered English editing at «MDPI's English editing service», and only after that, we submitted the edited manuscript to «Microorganisms». As you found the MDPI's English editing service inadequate, we contacted them again with your feedback, and manuscript was reedited. We hope that now the English language and style are fine
2) Line 74: The abbreviation “FT” must be explained
FT-agar and FT-broth are not abbreviations, but the names of commercial nutrient media for cultivation and isolation of the tularemia microbe, produced by SRCAMB and registered in Russia as a medical product, registration number ФСР2007/00899
3) Line 98: It must specific the subsp to which 15 NIIEG cells belong to (F. tularensis subsp. holarctica)
Done
4) Line 105: It must be emphasized that the three subspecies of F. tularensis were used for challenge
Done
5) Line 118, and tables 2 and 3: “,” must be changed to “.”. The decimals in English are expressed using “.” Instead of “,”
Done
6) Lines 137, Lines 138, 155 and 177: The first decimal must appear when writing this percentages
Fixed
7) Lines 164 and 165: These percentages does not match with those showing in Table 1. Please, edit them
We fixed this confusing error
8) Line 183: “Figure 5” must appear at the end of this paragraph
“Figure 5” even in the original version of the submitted manuscript appeared at the end of this paragraph. Due to its size, it did not fit on the same page as the text, but I am sure that in the printed version of the article the picture will be located next to the text.
9) Lines 207 and 225: The terms “conclusion” and “conclude” respectively should not be used because we still are in “Results” section
Fixed
10) Line 230: The word “text” has to be deleted in this context
Done. I apologize for a such negligence
11) Line 232: Please, write “guinea pigs” instead of “pigs” for avoinding confusions
Fixed
12) Line 239: Please, delete “p… > 0,5” because no significant differences are not necessary in the text 2
Done
13) Lines 257 to 445: Discussion section is too long. It must be reduced approximately one page. On the other hand, results are repeated very extensively in “discussion” section. We propose combine “results” and “discussion” sections for avoiding these repeats if possible. If not, results have to be minimized at the most
We have taken into account your remark. The «Discussion» section has been shortened. The «Results» section has also been slightly shortened to avoid repeats.
14) Line 260, 266 and 282: A reference must be included at the end of “…is also true for subsp. mediaasiatica”, “… this protection” and “…laboratory animals”.
We added relevant links
15) Lines 307 to 310: This statement must be supported by previous studies. If not, it must be deleted
To my regret, I did not understand if this comment refers to the submitted version of the manuscript or to the version that was returned to me as "Manuscript for Revisions", as they have slightly different line numbering. I took it upon myself to decide what you had in mind the original version and the phrase «At the same time, the formation of post-vaccination immunity is a dynamic process; immunity is not formed at once, and within a certain time, it decreases until it disappears completely»
We rewrote this phrase and added two references
(lines 260-261 in revised manuscript)
16) Lines 409 and 410: This statement must be founded in something
I guess you mean phrase « Most likely, the virulence and pathogenicity of subsp. mediasiatica is even lower for humans than for guinea pigs.» To change the extent of the allegation we rewrote this phrase
(lines 362-364 in revised manuscript)
17) Lines 438 to 440: How can the authors assert this if as they affirm in other part of the manuscript, there are no attenuated strains of F. tularensis subsp. mediasiatica?
We apologize for the mistranslations which led to misrepresentation of this phrase. We rewrote it (lines 390-392 in revised manuscript)
Round 2
Reviewer 1 Report
The manuscript has been improved.